# Mapping Community Priorities for Local Medical Centers: An Importance-Performance Analysis Study of Residents’ Perceptions in Large Cities, Non-Large Cities, and Rural Areas in South Korea

**DOI:** 10.3390/healthcare13192513

**Published:** 2025-10-03

**Authors:** Hana Jeong, Jaehee Seo, Eunyoung Chung

**Affiliations:** 1Department of Healthcare Program Development, Chungcheongnam-do Public Health Policy Institute, Daejeon 35015, Republic of Korea; jhn@chnpi.or.kr; 2Public Health and Medical Services Office, Chungnam National University Sejong Hospital, Sejong 30099, Republic of Korea; jehui.seo@gmail.com; 3Chungcheongnam-do Public Health Policy Institute, Daejeon 35015, Republic of Korea

**Keywords:** Local Medical Center, regional health equity, policy priorities, importance-performance analysis, Korea

## Abstract

**Background:** Policymakers in Korea are calling for Local Medical Centers (LMCs) to address regional healthcare disparities by expanding their roles beyond safety-net functions yet often overlook local community perspectives. **Methods:** Face-to-face survey data collected in 2022 from 2057 adults residing in Chungcheongnam-do were analyzed in this study, using Importance–Performance Analysis to assess how residents of large cities, non-large cities, and rural areas prioritize nine LMC functions. **Results:** While all valued public health policy and infectious disease control amid COVID-19, notable regional variations appeared: non-large city residents prioritized unmet healthcare needs and operational efficiency, rural respondents emphasized post-discharge care coordination due to aging and chronic disease, and large city residents focused on safety-net roles. Staff training and medical innovation ranked lowest across regions. **Conclusions:** The results highlight the inadequacy of one-size-fits-all policies and the importance of regionally tailored, resident-informed strategies for equitable public health in Korea.

## 1. Introduction

Addressing geographical disparities in healthcare remains a defining challenge for health systems worldwide [1]. Among these, the gap in medical resource distribution and service accessibility between urban and rural populations is both persistent and well-documented [2]. Statistical trends across decades confirm that communities outside major population centers continue to face significant barriers to accessing necessary care, resulting in uneven health outcomes and increased vulnerability among certain groups [3,4]. In South Korea (hereafter, Korea), this issue extends beyond the conventional rural–urban divide. The country exhibits a marked structural concentration of healthcare infrastructure and specialized facilities in major metropolitan areas, including the Seoul capital region and several large provincial cities, which has entrenched patterns of distorted healthcare utilization and imbalanced accessibility across regions [5,6].

Recognizing the enduring nature of these challenges, leading international organizations such as the World Health Organization have advocated for leveraging public healthcare infrastructure as a means of narrowing regional gaps [2]. Practical experiences in various national contexts suggest that strengthening public hospitals and integrating them into broader local service networks can play a pivotal role in securing equitable healthcare for populations regardless of their geographic location [2,7]. 

Reflecting this global perspective, Korea has developed a network of Local Medical Centers (LMCs, jibanguiryowon), which are public general hospitals designated by central or local governments to ensure essential healthcare services in underserved areas [8]. Traditionally, LMCs have functioned as regional safety nets, focusing on basic and emergency care to reduce health disparities. Since 2018, however, national policy has increasingly positioned LMCs as central hubs within each healthcare service area, assigning them expanded responsibilities such as coordination of regional healthcare delivery, leadership in public health initiatives, and strengthening primary care, especially where private provision is limited. These policy shifts underscore the evolving, central role of LMCs in Korea’s regional healthcare system [9,10,11]. 

Nevertheless, simply imposing the same institutional functions on every region risks overlooking genuine local diversity [12]. Not only does the contrast between urban and rural settings shape healthcare needs and access, but considerable variation also exists within urban areas themselves [12,13]. Notably, the healthcare landscape in Korea demonstrates acute heterogeneity between large metropolitan regions—such as Seoul and other principal cities—and smaller regional urban centers. Major metropolitan areas are distinguished by disproportionate concentrations of hospital capacity, highly specialized clinical services, and elevated patient volumes, whereas small and mid-sized cities are characteristically limited in both infrastructure and availability of medical specialists [12,13,14,15,16]. This spatial bifurcation in resource allocation and health service utilization has garnered sustained attention in policy analysis, thereby necessitating the development of differentiated interventions that account not only for urban–rural divides but also for the nuanced disparities among city types within the national context [5,6].

Against this backdrop, there exists an urgent scholarly imperative to delineate the perspectives and priorities of community residents through methodologically rigorous, large-scale empirical inquiry [17]. The prevailing reliance on data derived from healthcare professionals and institutional insiders has yielded a constrained understanding of community health needs and masked potential heterogeneity in local priorities for LMC functions across diverse geographic regions [18,19,20]. As highlighted by recent systematic reviews, there remains a critical need for studies employing community-based participatory research (CBPR) approaches to systematically capture residents’ views and priorities regarding the policy functions of public hospitals [21,22,23]. 

To address these needs, the present study focuses on Chungcheongnam-do, a province that exemplifies the diverse spectrum of metropolitan, small-to-medium-sized, and rural districts, offering a relevant context for analyzing regional variations in expectations regarding the roles of LMCs. Employing Importance–Performance Analysis, this research systematically examines which LMC functions residents perceive as most essential, with particular attention to differences across urbanization levels. The findings are expected to inform the development of refined, regionally responsive public healthcare policies that are more closely aligned with community-specific priorities. 

## 2. Materials and Methods

### 2.1. Data Source and Study Population

The present study was based on secondary survey data collected to represent the adult resident population (≥19 years) of Chungcheongnam-do, Korea. According to methodological documentation provided by the data source, quota sampling by sex, age group, and region (si/gun/gu) was employed, with Cheonan-si further divided into Dongnam-gu and Seobuk-gu. Sampling quotas were determined using the November 2021 Resident Registration Population Statistics, and a prioritized proportional allocation procedure was applied to ensure a minimum sample size in each quota cell, with the remaining sample allocated proportionally to population size.

Data collection was conducted by trained interviewers from Gallup Korea between 3 January and 28 January 2022, under commission from the Chungcheongnam-do Public Health Policy Institute (CHNPI). One-on-one, face-to-face interviews yielded a total of 2057 valid responses. The survey documentation reported that this sampling design corresponded to a margin of error of ±2.2 percentage points at the 95% confidence level, reflecting the intended sampling precision of the dataset.

The dataset included sample adjustment weights constructed by the data provider to align the achieved sample distribution with the underlying population structure of Chungcheongnam-do. No missing values were identified for the study variables, and the full dataset was used in the analysis. Access to the dataset required completion of the formal approval process mandated by the data provider, and the methodological details reported here were obtained from internal documentation supplied during this process. The study protocol was reviewed and approved for exemption by the Institutional Review Board of the National Institute for Bioethics Policy (IRB No. P01-202210-01-008).

### 2.2. Measures

#### 2.2.1. Importance–Performance Analysis Instrumentation and Measurement

This study performed secondary analysis utilizing the Importance–Performance Analysis (IPA) items embedded in the 2022 Chungcheongnam-do Public Healthcare Survey, which assessed residents’ perceptions of the importance and performance of nine core functions of LMCs. The original survey instrument was developed by the CHNPI, informed by the essential mandates of LMCs [24] and their designated responsibilities as outlined by relevant government policies [8,9,11]. Content validity was ensured during the original instrument development process by an expert panel comprising 10 public healthcare professionals and practitioners, who reviewed the draft items for relevance, clarity, and comprehensiveness.

For the present analysis, internal consistency of the IPA measurement scales was evaluated using Cronbach’s alpha, based on the full survey data. The Importance scale yielded an alpha of 0.981 and the Performance scale an alpha of 0.977, both demonstrating excellent reliability exceeding accepted thresholds.

Respondents had rated the perceived importance and performance of each LMC function using an 11-point scale ranging from 0 to 10, where 0 indicated “not at all important/performed,” 5 indicated “moderately important/performed,” and 10 indicated “extremely important/performed.” Higher scores reflected greater perceived importance or performance. The nine LMC functions included in the survey are presented in Table 1.

#### 2.2.2. Residential Area

Residential area was first classified according to the Korean administrative district system, with “si” (city) representing urban areas and “gun” (county) representing rural areas. In line with the Enforcement Decree of Local Autonomy Act, we further reflected the heterogeneity of urban areas by subdividing cities based on population size and healthcare infrastructure. Specifically, Cheonan—whose population exceeds 500,000—was designated a large city, consistent with thresholds commonly employed in national health policy. All other “si” were grouped as non-large cities due to their smaller populations and more limited resources. Accordingly, the 15 administrative districts of Chungcheongnam-do were categorized as follows: large city (Cheonan-si), non-large cities (Gongju-si, Boryeong-si, Asan-si, Seosan-si, Nonsan-si, Gyeryong-si, Dangjin-si), and rural areas (Geumsan-gun, Buyeo-gun, Seocheon-gun, Cheongyang-gun, Hongseong-gun, Yesan-gun, Taean-gun). 

This multistep classification—initially distinguishing urban and rural regions and then differentiating between large and mid-sized cities within the urban category—reflects both administrative conventions and practical differences in healthcare resource distribution. By adopting this approach, the study aligns with established policy and research practice and enables meaningful regional comparisons within Chungcheongnam-do. 

#### 2.2.3. Sociodemographic Variables

Four sociodemographic variables were included in the analysis: sex, age, educational attainment, and type of health insurance. Age was categorized into five groups: 19~29, 30~39, 40~49, 50~59, and 60 years or older. Educational attainment was categorized as middle school graduate or below, high school graduate, and college graduate or above, based on the distribution of the population. Type of health insurance was classified as either National Health Insurance (including both employee and self-employed subscribers) or Medical Aid (including both type I and II recipients as well as persons of national merit). 

#### 2.2.4. Health Status Variables

Two health status variables were included in the analysis: self-rated health status and the number of chronic diseases. Self-rated health status was measured using a five-point Likert scale and subsequently recategorized into three groups: poor health (“very poor” or “poor”), fair (“fair”), and good health (“good” or “very good”). The number of chronic diseases was assessed by asking participants how many chronic diseases they were currently diagnosed with, and responses were categorized as follows: none, one, or two or more.

### 2.3. Data Analysis

All statistical analyses were conducted using IBM SPSS Statistics, version 27.0 (IBM Corp., Armonk, NY, USA) with sampling weights applied. Differences in sociodemographic characteristics and health status variables across residential areas were examined using the chi-square test, with statistical significance set at *p* < 0.05 (two-sided). Residents’ perceptions of the core functions of Local Medical Centers (LMCs) were assessed by calculating the mean and standard deviation of ratings for both importance and performance, with 95% confidence intervals for the mean estimates reported. Between-area differences were examined using Welch’s ANOVA, which is robust to heterogeneity of variances and unequal group sizes, followed by the Games–Howell post hoc test for pairwise comparisons. Effect sizes were reported as *partial η^2^* for omnibus tests.

An importance–performance analysis (IPA) was then conducted to identify priority areas for the core functions of LMCs from the perspective of community residents. For the IPA, each function’s mean scores of perceived performance (*X*-axis) and importance (*Y*-axis) were plotted on a two-dimensional matrix, with quadrant boundaries set by the group-specific mean values of importance and performance for each residential area (i.e., large city, non-large cities, and rural area). This matrix distinguished four quadrants: “Keep Up the Good Work” (high importance, high performance), “Concentrate Here” (high importance, low performance), “Low Priority” (low importance, low performance), and “Possible Overkill” (low importance, high performance), which guided recommendations on which LMC functions should be maintained, improved, deprioritized, or possibly reallocated (Figure 1). All analyses, including the IPA, were stratified by residential area: large city, non-large city, and rural area.

To address the lack of uncertainty in traditional IPA, a bootstrap analysis (1000 replications) was performed to generate confidence intervals for each function and for the crosshair, thereby visualizing the stability of quadrant assignments. This procedure enabled us to evaluate whether each function’s placement remained stable within the 95% confidence intervals, thus providing a more robust interpretation of the IPA results (Figure 2).

## 3. Results

### 3.1. Sociodemographic and Health Status Profiles of Participants Across Residential Areas

Significant differences were observed across residential areas in sociodemographic and health status variables (Table 2). In rural areas, older adults aged 60 years and above accounted for 49% of respondents, whereas only 22.1% of respondents in the large city were in this age group. Educational attainment also differed across residential areas: in rural areas, 42.5% had completed middle school or less, while in the two urban areas, around 38% were college graduates or higher. The proportion of participants with at least a high school education was highest in the large city (84.6%), followed by non-large cities (73.7%), and lowest in rural areas (57.4%). 

In terms of health insurance type, there were no significant differences among residential areas; over 97% of respondents in all three groups were enrolled in National Health Insurance. Health status variables also varied significantly by residential area. In rural areas, 15.8% of participants considered their own health as “poor”, compared to 11.9% in non-large cities and 5.3% in the large city. The proportion of participants with at least one chronic disease exceeded half in rural areas, reaching 53.2%. This rate was also relatively high in non-large cities (44.4%), while only 25.3% of respondents in the large city reported having at least one chronic disease. Furthermore, the proportion of participants diagnosed with two or more chronic conditions was highest in rural areas (29%), followed by non-large cities (22.6%), and lowest in the large city (9.9%).

### 3.2. Regional Patterns in the Perceived Importance and Performance of Local Medical Center Roles

Table 3 presents the mean scores of perceived importance and actual performance for LMC roles across residential areas. Among the three residential areas (large city, non-large city, rural area), non-large city residents reported the highest mean scores for both perceived importance and performance of LMC functions. Rural area residents showed intermediate values, while large city residents reported the lowest mean scores in both dimensions. The average importance score across functions was 8.14 in non-large cities, 7.69 in rural areas, and 7.51 in large cities. The corresponding performance scores were 7.24 in non-large cities, 6.39 in rural areas, and 5.96 in large cities. The mean gap between importance and performance was largest in large cities (1.55), intermediate in rural areas (1.30), and smallest in non-large cities (0.90). Statistically significant differences in the importance–performance gap were observed across regions (*p* < 0.001), with full summary statistics provided in Appendix A.

The mean importance scores for individual functions ranged from 7.33 to 8.27, while mean performance scores ranged from 5.82 to 7.44, depending on role and residential area. Across all residential areas, Infectious Disease Control (Function G) and Public Health Policy Implementation (Function F) consistently occupied the highest ranks in both perceived importance and actual performance of LMC functions. Notably, among mid-sized city residents, Operational Efficiency (Function J) was identified as the second most important function, distinguishing this group from other residential areas. Medical Knowledge Innovation (Function B) received the lowest scores for both importance and performance regardless of residential area. The second lowest rankings, however, differed regionally: Staff Professional Training (Function A) in non-large cities and rural areas, and High-Quality Healthcare for Residents (Function C) in large cities. 

### 3.3. Final Priorities Determined by Importance Performance Analysis

Figure 2 displays the IPA matrix results for the nine functions of LMCs across large cities, non-large cities, and rural areas, categorizing each function into quadrants reflecting combined levels of perceived importance and actual performance. 

In large cities, Function E (Care for Vulnerable Populations), F (Public Health Policy Implementation), and G (Infectious Disease Control) were placed in Quadrant I, indicating consistently high importance and strong performance; these functions are suggested to be maintained. No functions were identified in Quadrant IV (“Possible Overkill,” Reduce), which indicates the absence of functions that are low in importance but high in performance in this analysis. The remaining functions—Function A (Staff Professional Training), B (Medical Knowledge Innovation), C (High-Quality Healthcare), D (Unmet or Essential Healthcare Services), H (Coordinated Post-Discharge Care), and J (Operational Efficiency)—were classified as “Low Priority” (Quadrant III). 

For non-large cities, Function F and G were likewise situated in Quadrant I, designating them for continued emphasis. However, Function C (High-Quality Care), D (Unmet or Essential Healthcare Services), and J (Operational Efficiency) were placed in Quadrant II, indicating that these are perceived as areas of high importance yet relatively low performance, thus requiring focused improvement. Only Function H appeared in Quadrant IV (“Possible Overkill, Reduce”), while Function A, B, and E remained in Quadrant III. 

In rural areas, Function C, E, F, G, and J occupied Quadrant I, evidencing concurrent high importance and high performance from respondents’ perspectives. Quadrant II included only Function H, highlighting Post-Discharge Care Coordination as the sole function needing prioritized improvement in this setting. Function A, B, and D were allocated to Quadrant III. 

The IPA matrix plots the average perceived performance (*X*-axis) and importance (*Y*-axis) of each function on a two-dimensional grid. The letters A to J represent the LMC functions described in Table 1. Quadrants are defined by each group’s mean importance and performance scores, as shown in Table 3. Both axes range from 0 to 10. Error bars, where shown, represent 95% confidence intervals estimated from 1000 bootstrap resamples. Quadrant definitions: I Quadrant = “Keep Up the Good Work” (high importance, high performance); II Quadrant = “Concentrate Here” (high importance, low performance); III Quadrant = “Low Priority” (low importance, low performance); IV Quadrant = “Possible Overkill” (low importance, high performance).

Overall, these results demonstrate that while maintenance of core public health functions (Function F and G) remains a priority across all regions, the specific roles necessitating concentrated improvement efforts differ by residential area. In non-large cities, High-Quality Care, Unmet or Essential Healthcare Services, and Operational Efficiency emerged as critical targets for enhancement, whereas in rural areas, Post-Discharge Care Coordination was uniquely prioritized. Large cities did not display specific high-priority deficits according to this IPA. Table 4 summarizes the functions located in each of the four quadrants according to residential area.

## 4. Discussion

This study provides a comprehensive examination of how residents from large cities, non-large cities, and rural areas assign relative priority to the diverse functions of LMCs. The findings reveal not only underlying commonalities, but also distinct contextual patterns shaped by each region’s unique healthcare landscape. 

Universally, Public Health Policy Implementation (Function F) remained firmly established as a foundational mission of public hospitals, while Infectious Disease Control (Function G) emerged as a top-tier expectation—particularly amplified by collective experience during the COVID-19 pandemic, which elevated public attentiveness to epidemic preparedness and response [25]. These dual priorities reflect both longstanding institutional mandates and a sharpened awareness of public health vulnerabilities in times of crisis. 

Differences across regions were equally pronounced. In non-large cities, residents attached heightened significance to Unmet or Essential Healthcare Services (Function D) and Operational Efficiency (Function J), pointing to persistent service deficits in areas such as emergencies, rehabilitative, and psychiatric care, coupled with an increasing demand for effective, transparent management [26,27]. These priorities were especially salient in contexts where the private healthcare sector is less developed, revealing elevated expectations for public institutions to fill crucial service gaps [28]. In rural areas, Coordinated Post-Discharge Care (Function H) surfaced as a particularly pressing concern—even in the face of infrastructure limitations comparable to or exceeding those in non-large cities. This intensive demand appears to be closely linked to rapid demographic aging and an associated rise in chronic disease burden, reinforcing the need for seamless continuity of care and integrated support systems during transitions from hospital to home [29]. Conversely, in large metropolitan centers, resident expectations remained anchored on traditional safety-net roles, while novel or specialized institutional functions received comparatively lower prioritization. Such attitudes may derive from the dense and diverse urban healthcare ecosystem, which reduces sole dependence on LMCs for specialized or ancillary services [30,31]. 

Across all study areas, roles encompassing Staff Professional Training (Function A) and Medical Knowledge Innovation (Function B) were persistently deemed least essential, both in importance and perceived execution. This suggests that communities currently view LMCs predominantly as care-delivery platforms, rather than as centers for professional advancement or research-based innovation—an outlook with potential implications for their long-term strategic development as called for in evolving health system policy [32,33]. Consistent with previous research, the aggregate results highlight areas meriting sustained policy attention toward more context-responsive strategies in LMC governance and resource allocation [34]. For large cities, future policy deliberations may benefit from closely examining conventional safety-net roles—particularly Functions E and F—and maintaining vigilance regarding Infectious Disease Control (Function G), as emphasized by recent pandemic experiences [35]. Efforts to broaden functional mandates for LMCs in metropolitan settings can be informed by systematic assessments of local provider capabilities and institutional niches, while recognizing the limits of cross-sectional observation.

In non-large urban environments, policy discussion may be guided by observed gaps in essential healthcare domains and opportunities to enhance operational efficiency via targeted institutional support and resource distribution. For rural settings, emergent evidence suggests the value of ongoing attention to robust post-discharge care integration to serve elderly and chronically unwell populations, aligning with national movements toward community-centered, integrated care models and reflecting demographic and epidemiological realities [7,36,37]. 

Taken together, these observational insights suggest the desirability of regionally tailored approaches to LMC development, with frameworks such as IPA offering avenues to identify actionable gaps and support more equitable improvements in public healthcare delivery. National mandates that are responsive to local priorities may contribute to increased adaptability and legitimacy of Korea’s regional health systems, although further longitudinal research is necessary to clarify causal pathways and implementation impacts [38].

Several limitations should be acknowledged in interpreting these findings. First, while this analysis emphasized external contextual determinants—such as urbanization level and population composition—it did not systematically incorporate the internal organizational features of LMCs, including staffing, resources, leadership, or historical performance. This omission may limit the explanatory power regarding performance variability between regions, as emerging evidence underscores the dynamic interplay between institutional capacity and external context [39,40]. In addition, grouping regions as large cities, non-large cities, and rural areas facilitated comparative analysis but may inadequately account for substantial intra-group heterogeneity in demographic profiles, resource endowments, and local healthcare needs. Second, the cross-sectional, self-reported study design introduces well-known risks of bias, including recall, social desirability, and residual confounding. Regarding response bias, responses collected in face-to-face surveys remain susceptible to participant bias, and the absence of longitudinal follow-up precludes any causal inference or examination of temporal trends. Finally, unmeasured confounders—ranging from local policy reforms to variations in private sector availability and pandemic-era shocks—may also have influenced the observed trends, and the specific timing of data collection (during population recovery from COVID-19) may further restrict generalizability. Accordingly, the results should be interpreted with caution, and future research would benefit from designs that integrate institutional-level data, address intra-group diversity more directly, and employ longitudinal or mixed-methods approaches to further elucidate the multi-level determinants of LMC performance and community expectations.

Despite these caveats, the present study offers several notable contributions to the public health systems literature. It is among the first to systematically delineate regional priorities across the functional spectrum of LMCs, directly incorporating resident perspectives that are often marginalized in top-down policy analyses. The detailed IPA-based methodology not only enables precise distinction of domains requiring maintenance or intervention but also provides a broadly applicable management and policy evaluation tool. The synthesis of regional findings enriches the ongoing discourse on health equity, underscoring both divergent and shared community needs. Ultimately, this research advances the case for resident-centered, data-informed governance of public hospitals—positioning Korea’s LMCs for greater system responsiveness, sustainability, and legitimacy. 

## 5. Conclusions

This study set out to examine whether the Korean government’s approach of assigning uniform functions to all LMCs effectively addresses regional healthcare equity and meets the lived realities of local communities. By systematically analyzing how residents in large cities, mid-sized cities, and rural areas prioritize LMC functions, we identified clear gaps between top-down policy intentions and the diverse needs and expectations of regional populations. 

Residents across all settings consistently valued LMC leadership in public health policy implementation and, particularly since the COVID-19 pandemic, infectious disease control. Nevertheless, pronounced regional differences emerged: non-large cities exhibited pressing needs for unmet or essential healthcare services and operational efficiency, while rural communities prioritized seamless post-discharge care in response to population aging and the rising burden of chronic disease. Large city residents, in contrast, maintained expectations for more traditional safety-net roles, likely reflecting the presence of a broader healthcare infrastructure.

These findings suggest that a one-size-fits-all policy for LMCs may not fully capture the heterogeneity of local healthcare contexts. For policies to be effective and equitable, greater attention to regional priorities and contextual diversity appears warranted.

Furthermore, the cross-sectional, self-reported nature of this study means that no causal inferences can be drawn; observed associations should be interpreted as indicative rather than definitive. The IPA framework applied here proved useful for identifying service gaps and informing potential priorities for resource allocation, but results require confirmation in longitudinal and mixed-methods studies.

By placing community perspectives at the center of this assessment, the study highlights the value of adaptive, evidence-informed governance for Korea’s local public health systems in an era of rapid change. Future research should examine whether the identified patterns persist as public health needs and policy environments evolve.

## Figures and Tables

**Figure 1 healthcare-13-02513-f001:**
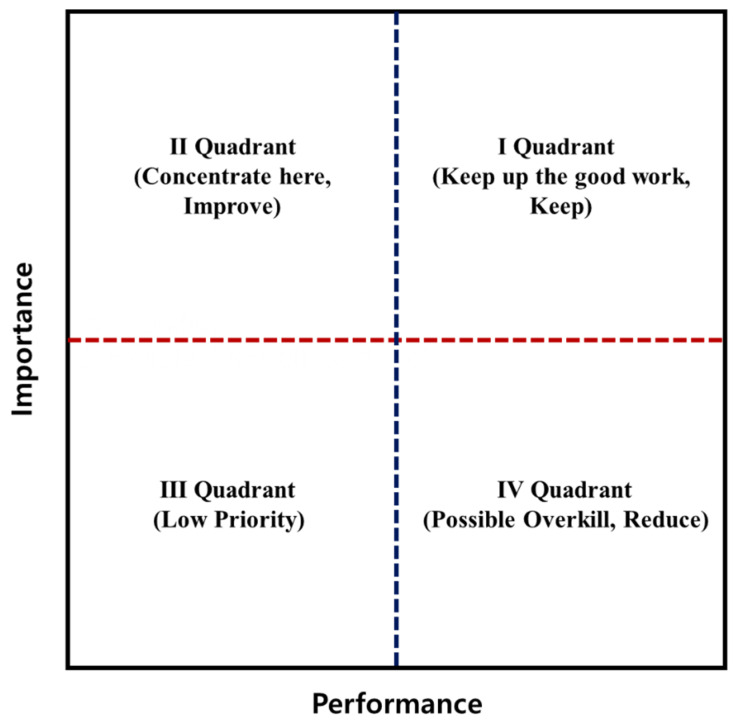
Importance-Performance Analysis (IPA) Matrix.

**Figure 2 healthcare-13-02513-f002:**
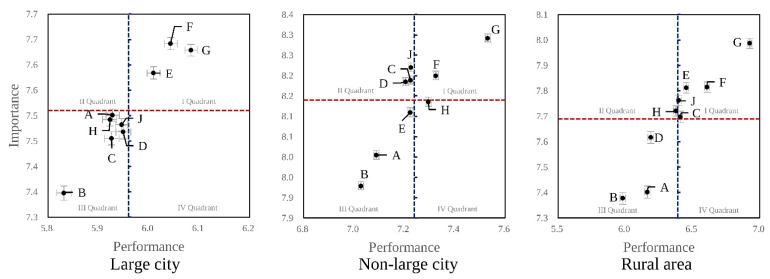
Importance-Performance Analysis matrix of nine core functions of Local Medical Centers by residential area (large, non-large city, rural).

**Table 1 healthcare-13-02513-t001:** Functions of Local Medical Centers Included in the IPA Items.

Function Label and Abbreviation		Description
A	Staff professional training	Offering professional development and training opportunities for medical staff
B	Medical knowledge innovation	Development of medical knowledge and innovation in disease treatment technologies
C	High-quality healthcare	Provision of high-quality healthcare for local residents
D	Unmet or essential healthcare services	Delivering unmet or essential healthcare services, including advanced emergencies, psychiatric, rehabilitation, maternal, and neonatal care, which are often avoided by private healthcare providers
E	Care for vulnerable populations	Ensuring access to medical care for vulnerable populations—such as the economically disadvantaged, women, the elderly, persons with disabilities, and those living in remote communities
F	Public health policy implementation	Implementation of public health policies and initiatives by the national government or the Chungcheongnam-do Provincial Government
G	Infectious disease control	Provision of a comprehensive range of services for the prevention, detection, management, and control of infectious diseases
H	Coordinated post-discharge care	Providing coordinated care for discharged patients by fostering close collaboration and referral networks among hospitals, clinics, and community welfare services
J	Operational efficiency	Proactive measures to advance operational efficiency and institutional competence

Note: The alphabetical sequence of labels omits the letter “I” to prevent confusion, as “I” is reserved for “Importance” scores in all analyses.

**Table 2 healthcare-13-02513-t002:** Sample Characteristics by Residential Area.

Variables	Categories	Total *n* (%)	Large City *n* (%)	Non-Large City *n* (%)	Rural Area *n* (%)	χ2	*p*
		2057 (100)	624 (30.3)	991 (48.2)	442 (21.5)		
Sex	Male	1049 (51.0)	318 (51.0)	511 (51.6)	220 (49.8)	0.4	0.822
	Female	1008 (49.0)	306 (49.0)	480 (48.4)	222 (50.2)		
Age (year)	19~29	229 (14.5)	118 (18.9)	136 (13.7)	45 (10.2)	97.7	<0.001
	30~39	301 (14.6)	119 (19.1)	144 (14.5)	38 (8.6)		
	40~49	377 (18.3)	129 (20.7)	188 (19.0)	60 (13.5)		
	50~59	394 (19.1)	120 (19.2)	191 (19.3)	83 (18.7)		
	≥60	687 (33.4)	138 (22.1)	332 (33.5)	217 (49.0)		
Educational attainment	≤Middle school	545 (26.5)	96 (15.4)	261 (26.4)	188 (42.5)	105.7	<0.001
	High school	775 (37.7)	292 (46.8)	348 (35.2)	135 (30.5)		
	≥College	736 (35.8)	236 (37.8)	381 (38.5)	119 (26.9)		
Type of health insurance	National Health Insurance	2001 (97.3)	606 (97.1)	962 (97.2)	433 (97.7)	0.5	0.792
	Medical aid	56 (2.7)	18 (2.9)	28 (2.8)	10 (2.3)		
Self-rated health status	Poor	221 (10.7)	33 (5.3)	288 (29.1)	70 (15.8)	46.9	<0.001
	Moderate	538 (26.1)	142 (22.7)	287 (29.8)	108 (24.4)		
	Good	1300 (63.1)	450 (72.0)	585 (59.0)	265 (59.8)		
Number of diagnosed chronic disease	0	1224 (59.5)	466 (74.7)	551 (55.6)	207 (46.8)	102.7	<0.001
	1	419 (20.4)	96 (15.4)	216 (21.8)	107 (24.2)		
	≥2	414 (20.1)	62 (9.9)	224 (22.6)	128 (29.0)		

**Table 3 healthcare-13-02513-t003:** Regional Differences in Residents’ Perceived Importance and Performance of the Functions of Local Medical Centers.

Rank	Importance						Performance					
	Large City (*n* = 624)		Non-Large City (*n* = 990)		Rural Area (*n* = 443)		Large City (*n* = 624)		Non-Large City (*n* = 990)		Rural Area (*n* = 443)	
	Function	M ± SD	Function	M ± SD	Function	M ± SD	Function	M ± SD	Function	M ± SD	Function	M ± SD
1st	F	7.64 ± 1.7	G	8.29 ± 1.5	G	7.99 ± 1.9	G	6.08 ± 1.4	G	7.53 ± 1.5	G	6.93 ± 1.6
2nd	G	7.63 ± 1.6	J	8.22 ± 1.5	F	7.81 ± 2.1	F	6.04 ± 1.6	F	7.32 ± 1.5	F	6.61 ± 1.8
3rd	E	7.58 ± 1.7	F	8.20 ± 1.5	E	7.81 ± 2.1	E	6.01 ± 1.6	H	7.30 ± 1.6	E	6.45 ± 1.9
4th	H	7.50 ± 1.7	C	8.19 ± 1.5	J	7.76 ± 2.1	D	5.95 ± 1.7	J	7.23 ± 1.6	C	6.41 ± 1.8
5th	A	7.49 ± 1.7	D	8.18 ± 1.6	H	7.72 ± 2.1	J	5.95 ± 1.6	C	7.22 ± 1.5	J	6.40 ± 1.8
6th	J	7.48 ± 1.7	H	8.14 ± 1.6	C	7.70 ± 2.3	H	5.93 ± 1.6	E	7.22 ± 1.6	H	6.38 ± 1.8
7th	D	7.47 ± 1.9	E	8.11 ± 1.6	D	7.62 ± 2.3	C	5.93 ± 1.7	D	7.20 ± 1.6	D	6.19 ± 2.1
8th	C	7.46 ± 1.8	A	8.00 ± 1.6	A	7.40 ± 2.3	A	5.92 ± 1.6	A	7.09 ± 1.6	A	6.16 ± 2.0
9th	B	7.35 ± 1.8	B	7.93 ± 1.7	B	7.38 ± 2.3	B	5.83 ± 1.8	B	7.03 ± 1.6	B	5.98 ± 2.0
Total mean		7.51 ± 1.6		8.14 ± 1.4		7.69 ± 2.1		5.96 ± 1.5		7.24 ± 1.4		6.39 ± 1.7

M = Mean; SD = Standard Deviation; The letters A to J represent the LMC functions described in Table 1 (Except for ‘I’); Cells corresponding to identical functions are presented with the same color for ease of visual comparison.

**Table 4 healthcare-13-02513-t004:** Comprehensive IPA Matrix of LMCs Across Residential Areas.

Quadrant of IPA Matrix	Large City	Non-Large City	Rural Area
I Quadrant (Keep up the good work, Keep)	E, F, G	F, G	C, E, F, G, J
II Quadrant (Concentrate here, Improve)	-	C, D, J	H
III Quadrant (Low Priority)	A, B, C, D, H, J	A, B, E	A, B, D
IV Quadrant (Possible Overkill, Reduce)	-	H	-

Note. The letters A to J represent the LMC functions described in Table 1.

## Data Availability

The data presented in this article are not permitted for use beyond research purposes and cannot be shared publicly. However, if access to the data is justified, requests may be addressed to the corresponding author, who will obtain the necessary approval from the institution before data sharing.

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
