# Peer review of "Mapping Community Priorities for Local Medical Centers: An Importance-Performance Analysis Study of Residents’ Perceptions in Large Cities, Non-Large Cities, and Rural Areas in South Korea"

_healthcare, 2025, doi:10.3390/healthcare13192513_

Round 1

Reviewer 1 Report

Comments and Suggestions for Authors

Reviewer Comments

This paper addresses the prioritization of geographical disparities in healthcare across LMC functions. The topic is within the scope of the journal and of great interest to its readership. The paper is well written and based on a well-considered study design. However, it would benefit from further revision before publication.

Specific Comments

  1. City Categories and Geographical Disparities
    • Please discuss the categorization of cities (rural, mid-size, and large) as a measure of geographical disparities. The large variations between mid-size cities and rural areas may not be fully captured when cities are grouped only by size. How can local diversity be better addressed?
  2. Margin of Error
    • Please explain the rationale for using a margin of error of 2.2. Why was this threshold selected?
  3. Missing Data and Sample Size
    • Please discuss the presence of missing data and describe how it was handled in the analysis.
    • Clarify how the sample size was determined.
  4. Potential Bias from Study Timing
    • Since the study was conducted in 2022, when the public was still recovering from the COVID-19 pandemic, please discuss potential bias. The findings may not fully reflect healthcare needs at other times. This is also apparent in the tables, where both policy implementation and infectious disease control rank among the top concerns across all cities.
  5. Limitations
    • Please move the Limitations section into the Discussion for better alignment with journal standards.

Reviewer 2 Report

Comments and Suggestions for Authors

Abstract

  • Add design year and mode (face-to-face) so readers can assess COVID-period context and mode effects here, not only in Methods.
  • Clarify whether “importance–performance” gaps were statistically tested across regions (currently framed descriptively only). Consider citing tests or CIs in one succinct sentence.
  • Consider quantifying at least one key effect (e.g., “importance–performance gap in large cities was 1.55 vs. 0.98 in mid-sized cities”).
  1. Introduction
  • The argument that “one-size-fits-all” is inadequate is persuasive, but cite Korean evidence more specifically (beyond general WHO guidance) where you claim within-urban heterogeneity and mid- vs large-city differences.
  • Tighten the “gap in literature” claim by specifying what prior studies did not do (e.g., lacked representative sampling, no IPA, no cross-regional stratification), then explicitly state how your study addresses each gap.
  • Minor: a few long sentences could be split for readability, and hyphenation artifacts (“differ-entiation”) appear—likely from PDF line breaks; fix in the source doc.
  1. Materials and Methods

2.1 Study design and sample

  1. Temporal inconsistency in sampling frame vs. fieldwork dates.
    You state quotas were based on November 2022 population stats but fieldwork occurred January 3–28, 2022—ten months earlier. That’s impossible and undermines reproducibility. Most likely the frame was November 2021 or the fieldwork was in January 2023. Please correct both year(s) consistently across manuscript, abstract, and any PRISMA/STROBE checklist.
  2. Primary vs. secondary data contradiction (ties to Ethics).
    Here you describe primary data collection by Gallup Korea via face-to-face interviews, yet the Ethics section later claims only secondary anonymized data were analyzed. Resolve this contradiction and ensure the IRB language matches the actual activity (see Ethics critique).

Methodological gaps & actionable fixes

  • Sampling implementation details missing: How were households/individuals selected (random route, Kish grid)? Response rate? Replacement rules? Add these.
  • For stratified quotas, report whether you computed post-stratification/raking weights to population margins. If not, justify; if yes, state how you used weights in all analyses. (No mention currently.)
  • Precision claim: “±2.2 pp at 95%” is global; clarify whether this is for simple random assumption or design-effect-adjusted (DEFF). Provide DEFF if available.

2.2 Measures

IPA questionnaire development

  • Missing Psychometrics (reliability and validity). Report Cronbach’s α (importance & performance subscales), test–retest (if available), and a brief EFA/CFA to justify unidimensional treatment of each function and cross-group measurement invariance (large vs mid-sized vs rural). Without this, cross-area mean comparisons may reflect measurement artifacts.

Item wording / construct clarity

  • Several items are double-barreled or overly broad (e.g., D bundles emergencies, psychiatric, rehab, maternal & neonatal; H mixes clinical and welfare integration; J is vague “operational efficiency”). Consider splitting or tightening items; add examples sparingly.
  • Scale anchors: 0–10 “not at all” to “extremely” but no intermediate descriptors; add at least 3–4 verbal anchors or show the exact question text in an Appendix.

Residential area

  • Classifying one large city (Cheonan) vs eight mid-sized cities vs seven rural counties yields very uneven, heterogeneous groups; this invites ecological confounding. Consider sensitivity analyses (e.g., multilevel models or excluding the single-city “large” category to check robustness).

Health status & utilization

  • LMC utilization conflates self and cohabiting family use into one binary—it’s ambiguous and mixes levels of analysis. Consider separate indicators or a weighted composite. Also define “past three years” relative to the actual fieldwork date (once corrected).

2.3 Data analysis

  • State that analyses used weights (if any), report design-based tests (Rao–Scott χ²) instead of simple χ² for complex samples, and add effect sizes (Cramér’s V for χ²; Cohen’s d for mean differences). Currently only p-values appear.
  • You set quadrant boundaries at total means across all functions; consider (a) scale-centered crosshairs or (b) region-specific crosshairs; at minimum, justify your choice and provide crosshair numeric values in the figure caption or appendix.
  1. Results

3.1 Sample profile

  • The p-value column mixes p-values (“0.8”) with χ² statistics plus asterisks (“93.4***”, “98.6***”). Present either test statistics with df and p-values, or p-values only—consistently. Also define the asterisks in the table note.
  • Report missingness (n, %) per variable. None shown.

3.2 Perceived importance & performance

  • Provide 95% CIs for means in Table 3 and state whether between-area differences were tested (ANOVA/linear models with robust SEs, or ordinal models if treating scores as ordinal). Currently appears descriptive only.
  • Since you highlight “mid-sized city residents rated J second most important,” consider testing that ranking difference formally (e.g., pairwise comparisons with multiplicity control).

3.3 IPA matrix

  • Text says “Figure 2 displays the IPA matrix,” but the caption that follows is “Figure 1.” Harmonize numbering.
  • Earlier you already had “Figure 1. Importance-Performance Analysis (IPA) Matrix” (generic). Avoid duplicate captions or renumber as Figure 1 (generic template) and Figure 2 (results by area).
  • “Table 54” is almost certainly “Table 4.” Renumber and cross-check all internal references.
  • Quadrant labels are swapped between the results text and Table 54: the figure caption defines Quadrant II = “Possible Overkill” and Quadrant IV = “Concentrate Here”, but the table lists Quadrant II as “Concentrate here” and Quadrant IV as “Possible Overkill”. Fix the table to match the canonical IPA definitions.

Methodological cautions to add

  • Emphasize that IPA uses sample means without uncertainty; add a brief sensitivity check (bootstrapped crosshair and point estimates; show whether quadrant assignments are stable within 95% CIs).
  1. Discussion
  • Avoid normative prescriptions that imply causality from cross-sectional perceptions; rephrase to “suggest policy attention” and flag observational limits.
  • Add a paragraph on potential response biases (acquiescence in mid-sized cities; mode effects from face-to-face interviews) and non-response bias (if response rate <60%). These are not discussed.
  • Acknowledge heterogeneity within groupings (eight different mid-sized cities, seven rural counties). Consider an appendix table showing within-group dispersion for transparency.
  1. Conclusions
  • Replace rhetorical statements with actionable, measurable recommendations (e.g., “For mid-sized cities, set a KPI to reduce the performance gap for D and J by ≥0.5 within 2 years, measured with annual resident surveys using the same instrument”).

Tables & Figures (general)

  • Ensure all figures include numeric crosshairs, axis ranges (0–10), and CIs if you adopt them. Current captions don’t show these values.
  • Standardize table notes: define asterisks, specify whether values are weighted.

Ethics, IRB, consent, and data availability (must fix)

Contradictions

  • Methods clearly describe primary data collection interviews, while the IRB and Consent sections state analysis of secondary anonymized data with no consent required. This is a serious compliance problem. Align the sections: either (a) this was secondary data (then remove the fieldwork description), or (b) this was primary data (then IRB language must reflect primary data with appropriate consent procedures).
  • If primary: briefly describe consent procedures, interviewer training to minimize coercion, and data protection steps. If secondary: replace Methods’ fieldwork details with a data source description (owner, collection dates, sampling, and your data-use agreement).
  • The “Data Availability” statement is restrictive (“cannot be shared publicly”) yet offers case-by-case access; consider depositing a de-identified, minimal dataset and the questionnaire in a repository to improve transparency/reproducibility.

References

  • Several web references include “accessed on 5–6 August 2025.” Confirm these are accurate relative to your final submission date; journal styles often require month day, year and stable URLs.
  • Ensure consistent romanization and English titles for Korean laws/reports, and verify that in-text citations (e.g., [6–9]) map to the correct items.

Reviewer 3 Report

Comments and Suggestions for Authors

This paper analyzed survey data from 2,057 participants in Chungcheongnam-do in South Korea, to assess how residents of large cities, mid-sized cities, and rural areas prioritize nine LMC functions using Importance–Performance Analysis. In lines 201-204 (page 6), the authors stated that most respondents and their cohabiting family members across all three groups had not used the LMCs in the past three years. The survey responses seem to be based on judgements that are not informed by their actual and more recent interactions with the LMCs. The absence of survey participants’ recent experiences and encounters with the LMCs casts doubts about the interpretation of their perceptions about the importance and performance of the core LMC functions. In this context, it is notable that the mean gap between importance and performance was largest in large cities (1.55) and the proportion of survey respondents with LMC experience was smallest in the large city (7.6%).

Secondly, the authors categorized urbanization as large and mid-sized cities. I am not sure if the specific province did not have any small cities at all or if the authors chose to leave out the residents of the small cities deliberately from their survey. This needs to be explained further.

Finally, importance and performance are not independent, and statistical measures such as correlation could be used to measure their association. The main problem is whether the authors provided convincing arguments to help readers understand the negative associations between importance and performance in the regions based on individual characteristics of survey respondents. Some functions such as staff professional training and medical knowledge and innovation mainly serve the needs of providers and may not be important to patients. High spending to enhance these functions without patients being unable to see their importance may lead to a wide divergence between performance and importance assessment by survey respondents. More discussions along these lines could further enrich the value and usefulness of the findings from this paper.

I am providing other comments below for the authors’ review.

Abstract, line 23: The statement “staff training and medical innovation ranked lowest across regions” is self-explanatory as these two functions mainly address the providers’ perspective.

Table 1: The abbreviation "I" was not used following the alphabetical order of labeling the functions. I think authors wanted to keep "I" for “Importance.” This could be explained at the outset.

Results, lines 181-204: I see that the rural area residents were mainly older individuals with lower level of education and higher number of comorbidities and higher self-rating of poor health. It is not surprising that they are more concerned about hospital discharges than antenatal or neonatal care that are of prime importance to young women in their reproductive age, who may be in majority in large cities. These factors could be considered to explain the assessment of importance of LMC functions by survey respondents in the three regions.

Tables 3 and 4: The abbreviations in the footnotes are already defined in Table 1 and may not be repeated again to avoid extra space requirement. I will, however, leave the decision to the authors if they feel that this is necessary to refresh the understanding of readers who are focusing on the tables.

3.3. Final Priorities Determined by Importance Performance analysis

There were errors in many places with reporting the right quadrants. For example, in the sentence “No functions were identified in Quadrant IV (“Concentrate here, Improve”), indicating the absence of functions simultaneously recognized as high priority but underperforming within this context,” (lines 256-258) Quadrant IV should be Quadrant II. Similarly, in lines 263-266, Quadrant IV should be replaced by Quadrant II. Again, in the sentence “Only Function H appeared in Quadrant II (“Possible Overkill, Reduce”)...” (lines 266-267), Quadrant II should be Quadrant IV, and in the sentence “Quadrant IV included only Function H, highlighting Post-Discharge Care Coordination as the sole function needing prioritized improvement in this setting” (lines 270-272), Quadrant IV should be Quadrant II.

Page 9: Figure 1 should be Figure 2. Also, the abbreviations were also defined in Table 1 and may be discarded from the footnotes as I suggested before. Additionally, within the text under this figure (lines 278-282), there is error in reporting Quadrant II and Quadrant IV.

Private providers are mentioned in several places in text (line 50, Table 1, line 339, line 387) without describing regional variations in their availability. They are also more costly than public health. Whether and how private provision of healthcare could complement public provision was not discussed in the paper.

Round 2

Reviewer 2 Report

Comments and Suggestions for Authors

Resolve the IPA crosshair inconsistency (Methods vs Fig. 2). Recommend adopting group-specific crosshairs to match your response and Figure 2 caption; edit L185–L191 accordingly.

Document the scale’s verbal anchors (0 / 5 / 10) in Methods 2.2.1.

Name the exact models actually used for between-area tests (e.g., Welch’s ANOVA and Games–Howell) and add effect sizes (partial η² for omnibus; Cohen’s d for pairwise) in text + Supplement labels (your response mentions this, but the manuscript currently just says “ANOVA”).

Figure caption completeness: add the axis range (0–10) and, if shown, that error bars reflect 95% CIs from 1,000 bootstrap resamples. You already describe bootstrap in Methods; mirror it in the caption

Reviewer 3 Report

Comments and Suggestions for Authors

Although I could not find the revised version of the paper, I saw the authors' responses to my previous comments. The authors have satisfactorily addressed all my comments, and I have no additional issues. Please ensure that the paper incorporates large and non-large cities and rural areas in the title. Also, expectations about performance and importance are often informed by prior experiences with the LMCs. The authors may want to clarify that this is not the case here.

Author Response

We appreciate the reviewer’s careful review and suggestions. As recommended, we carefully re-examined the entire revised manuscript and confirmed that all relevant references to regional categories now use “non-large city” terminology as appropriate, both in the title and throughout the text.
Additionally, we thank the reviewer for highlighting the potential influence of prior experiences with LMCs on respondents’ perceptions. We will consider addressing the effect of respondents’ prior interactions with LMCs as an important variable in future research.
Thank you again for your thoughtful comments, which helped improve the clarity and scientific value of our work.